# Latent TB Infection, Vitamin D Status and COVID-19 Severity in Mongolian Patients

**DOI:** 10.3390/nu15183979

**Published:** 2023-09-14

**Authors:** Davaasambuu Ganmaa, Tserendorj Chinbayar, Polyna Khudaykov, Erdenebileg Nasantogtoh, Sukhbaatar Ariunbuyan, Tserenkhuu Enkhtsetseg, Ganbold Sarangua, Andrew Chan, Dalkh Tserendagva

**Affiliations:** 1Channing Division of Network Medicine, Department of Medicine, Brigham and Women’s Hospital, Harvard Medical School, Boston, MA 02115, USA; 2Department of Nutrition, Harvard T.H. Chan School of Public Health, Boston, MA 02115, USA; 3National Center for Communicable Disease, Ulaanbaatar 13335, Mongolia; chinbayar0423@yahoo.com (T.C.); saranguag@yahoo.com (G.S.); 4Sage Therapeutics, Cambridge, MA 02142, USA; polinadr@gmail.com; 5National Center for Maternal and Child Health, Ulaanbaatar 16060, Mongolia; nasantogtox.e@gmail.com; 6Division of Oral and Maxillofacial Oncology and Surgical Sciences, Graduate School of Dentistry, Tohoku University, Sendai 980-8575, Japan; ariunbuyan@tohoku.ac.jp; 7Laboratory of Biomedical Engineering for Cancer, Graduate School of Biomedical Engineering, Tohoku University, Sendai 980-8575, Japan; 8Biomedical Engineering Cancer Research Center, Graduate School of Biomedical Engineering, Tohoku University, Sendai 980-8575, Japan; 9Mongolian Health Initiative (MHI), Ulaanbaatar 14210, Mongolia; enkhtsetseg.mhi@gmail.com; 10Department of Medicine, Brigham and Women’s Hospital, Harvard Medical School, Boston, MA 02115, USA; achan@mgh.harvard.edu; 11International School of Mongolian Medicine, Mongolian National University of Medical Sciences, Ulaanbaatar 14210, Mongolia; tserendagva@mnums.edu.mn

**Keywords:** tuberculosis, BCG vaccination, vitamin D, COVID-19 severity, Mongolia

## Abstract

We aimed to determine potential risk factors for COVID-19 severity including serum vitamin D levels and latent TB infection among Mongolian inpatients diagnosed with COVID-19, and to study the effects of disease complications and treatment outcomes. This study included patients admitted to the Mongolian National Center for Communicable Disease, a main referral center for infectious disease in Mongolia, with COVID-19 ascertained by a positive PCR test. Patients’ demographic, clinical, and laboratory data were analyzed. Of the 270 patients enrolled, 125 (46%) had mild-to-moderate illness, 86 (32%) had severe illness, and 59 (22%) had critical illness. Ten (91%) of the 11 patients who had active TB were hospitalized with severe or critical COVID-19, suggesting that they had a higher risk of falling into the severe category (OR = 10.6 [1.2; 92.0] 95% CI). Severe vitamin D deficiency (25(OH)D < 10 ng/mL) was present in 32% of the patients, but was not significantly associated with the severity of illness (*p* = 0.65). Older age, being male, having active TB and/or COPD were associated with greater COVID-19 severity, whereas a history of COVID-19 vaccination and the presence of a BCG vaccination scar were protective in terms of disease severity.

## 1. Introduction

Mongolia is located in Central Asia, between China and Russia, with a population of approximately 3.4 million people, of whom 1.2 million (39%) reside in the capital city of Ulaanbaatar. Mongols have an extraordinarily high prevalence of vitamin D deficiency, owing to Mongolia’s high latitude, cold weather, and a lack of vitamin D in the food supply. We have conducted a nationwide survey in urban and rural areas of Mongolia and found that the fraction of the adult population with deficient or inadequate serum 25(OH)D concentrations (<20 ng/mL) was 42% in summer and 100% in winter [1].

Some research suggests that vitamin D metabolites support both innate and acquired immune responses to respiratory viruses and bacteria, and may dampen inflammatory responses [2,3]. Both TB infection and severe acute respiratory syndrome-coronavirus 2 infection (SARS-CoV-2; and coronavirus disease 2019 (COVID-19)) are affected by innate and acquired immunity. Vitamin D, in the form of calcitriol, has a strong immunomodulatory effect in vivo, and in human and animal studies [4]. Clinical evidence suggests that SARS-CoV-2 predisposes patients to TB infection or may lead to reactivation of latent disease [5]. Similarly, underlying TB can worsen COVID-19 [6].

Only limited, if any, direct treatment options are available for COVID-19 so immunomodulatory approaches such as vitamin D supplementation, are appealing. Patients with severe COVID-19 who also had tuberculosis, have been reported to show increased inflammatory markers, increased mortality, and a higher rate of vitamin D deficiency [7,8]. Epidemiological studies have suggested that vitamin D deficiency might be a risk factor for severe COVID-19 and for TB [9,10]. Since these two diseases show adverse synergy in individual patients, there is even more reason to consider the possibility that vitamin D supplementation may mitigate severity and mortality in SARS-CoV-2 and COVID-19.

In Mongolia, there is a high prevalence of both vitamin D deficiency and tuberculosis [8]. Since the COVID-19 outbreak, vitamin D supplementation has increased dramatically in the general population (personal communication). However, no research studies have investigated the possible relationship between vitamin D status, tuberculosis (TB) and COVID-19. Therefore, we aimed to determine potential risk factors for COVID-19 severity including serum vitamin D levels and latent TB infection among Mongolian inpatients diagnosed with COVID-19, and to study their relationships to disease complications and treatment outcomes.

## 2. Methods and Materials

### 2.1. Participants

A definite COVID-19 case was defined by a positive PCR with clinical signs consistent with COVID-19. We have specified no exclusion criteria. All participants who were treated as inpatients for COVID-19 at the Mongolian National Centre for Communicable Disease (NCCD) were included. The study population is relatively homogenous with regard to ethnic background (predominantly Khalkh Mongol) and age (predominantly adult). Participants were enrolled on the first day of hospitalization. Clinical signs and symptoms were assessed at least daily by physical examination and questionnaire. Participants were followed from admission to discharge.

### 2.2. Data Collection and Measurements

Physicians and medical residents at NCCD collected information from each participant’s medical chart: age, sex, highest education level attained, type of residence, monthly household income, home ownership, active smoking, alcohol use, presence of an index case of pulmonary TB living in the household, medical history, use of vitamin D supplementation, presence of a BCG vaccination scar, medical comorbidities, history of COVID-19 vaccination, and anthropometric measurements. Body mass index (BMI) was calculated using the formula BMI = weight (kg)/height (m^2^). Disease severity was evaluated by WHO criteria at the time of admission as: mild, moderate (non-pneumonia and mild pneumonia), severe, and critical. Disease complications such as respiratory collapse, septic shock, with or without multiple organ failure (kidney, liver, and heart), coagulation dysfunction and/or cytokine storm, duration of individual symptoms, time to intensive care unit (ICU) admission, and time to discharge were all recorded. The specimens analyzed in this study were collected from participants during routine tests and procedures conducted as part of the local standard of care for COVID-19 patients in Mongolia. As allowed by the government of Mongolia’s national emergency declaration, the clinical standard of care provided allowances for storage of biochemical samples for research purposes. The following variables were measured each week during the admission: oxygen saturation, prothrombin time, coagulator factors, angiotensin-converting enzyme (ACE), albumin, serum ferritin, serum creatinine, procalcitonin, troponin, viral RNA, interleukin-6 (IL-6), and C-reactive protein (CRP). Inflammatory markers such as C-reactive protein (CRP, CRP-CLIA kit), Ferritin (Fer, Ferritin-CLIA kit), D-Dimer (D-Dimer-CLIA kit), interleikin-6 (IL-6), procalcitonin (PCT) were measured using a benchtop analyzer, the Maglumi 800 (Snibe Co., Ltd., Shenzhen, China), according to the manufacturer’s instructions. Fibrinogen, activated partial thromboplastin clotting time (aPTT), thrombin time (TT), prothrombin time (PT), white blood cells (WBC), hematocrit (HCT), neutrophils (NEUT), lymphocytes (LYMPH), monocytes (MON), serum creatinine (s-Cr) were measured using STA Compact (Diagnosto Stago, Asnières sur Seine CEDEX, France) and Sysmex XN550 (Sysmex Corporation, Kobe, Japan) according to the manufacturer’s instructions; and lymphocyte to monocyte ratio (LMR) and international normalized ratio (INR) were calculated. Serum 25(OH)D concentrations were determined using an enzyme-linked fluorescent assay (VIDAS 25OH Vitamin D total, Biomerieux, Marcy-l’Étoile, France). Total CV was 6% and the limit of quantitation (LOQ) was 8.1 ng/mL. The QuantiFERON-TB Gold (QFT-G) assay was performed according to manufacturer’s instructions at the Global Laboratory, Ulaanbaatar, Mongolia, which we have previously validated by participation in the QuantiFERON Quality Assurance Program of the Royal College of Pathologists of Australasia.

This study was approved by the Harvard SPH IRB and Mongolian Ministry of Health IRB (IRB # 20-1441).

### 2.3. Statistical Analysis

For the descriptive statistics, continuous data were summarized using means and standard deviations (sd) and categorical data were summarized using the number and percent of subjects for each category, where appropriate. The analysis was conducted using R (RStudio Team (2020). RStudio: Integrated Development for R. RStudio, PBC, Boston, MA, USA, URL: http://www.rstudio.com/, accessed on 30 June 2023).

The primary outcome, COVID-19 disease severity, was analyzed as a binary variable, which takes value 0 if the status is mild or moderate and “1” if the status is severe/critical. A logistic regression was used to assess the effect of risk factors on the disease severity. The odds ratio (OR) along with 95% CI were reported. Univariable models were used to assess the risk of each variable separately and then a multivariable model was constructed using all variables that had *p*-value < 0.1 in the univariable models. A similar analysis was conducted for the evaluation of the relationship of biomarkers on disease severity.

The duration of hospitalization was analyzed as time-to-event (discharge) using a multivariable Cox proportional hazard model for the following risk factors: COVID-19 severity status at admission, age, sex, comorbidities (yes/no), body mass index, smoking status (yes/no) and vitamin D deficiency. Hazard ratio (HR) and 95% confidence intervals were reported.

In all analyses, the vitamin D deficiency group was defined as serum 25(OH)D concentration <10 ng/mL. This threshold was pre-specified, based on findings of a previous study that reported susceptibility to M. tuberculosis infection to be increased below this cut-off [11]. Two sensitivity analyses were conducted after (1) applying a natural logarithm transformation to 25(OH)D, and (2) excluding observations below the limit of detection. Neither analysis produced significantly different parameter estimates or associated *p* values, so results of these analyses are omitted for simplicity.

## 3. Results

The study population was homogenous with regard to ethnic backgrounds (predominantly Khalkh Mongol). Of the 270 patients, 125 (46%) had mild-to-moderate illness, 86 (32%) had severe illness, and 59 (22%) had critical illness. The mean age of the COVID-19 patients was 56 years (sd 17.6). One hundred and sixty seven of them (62%) were women (Table 1). Two hundred and forty six (91%) of them had been vaccinated against COVID-19, among whom 101 (37%) had two doses of vaccine, 109 (40%) had been vaccinated with boosting dose III and 16 (6%) had been vaccinated with boosting dose IV (Table 1).

Three of the 270 patients died, giving an overall case fatality rate of 1%. Increased rates of comorbidity (chronic obstructive pulmonary disease, COPD, cardiovascular disease, CVD, parathyroid condition, sarcoidosis and sleep apnea) were observed more frequently in the severe and critical groups. Older age, being male, CRP, and the thrombin time (TT) were significantly associated with the severe and critical illness (Table 2 and Table 3). Histories of COVID-19 vaccination and BCG vaccination were associated with a lower risk of being classified as having severe or critical disease. Severe vitamin D deficiency (defined as 25(OH)D < 10 ng/mL) was present in 32% of the patients. However, serum 25(OH)D concentrations did not differ significantly between the patients whose illnesses were classified as mild or moderate compared to those whose illnesses were classified as severe or critical (*p* = 0.65). The time to discharge was associated with COVID-19 severity symptoms at hospitalization (HR = 0.67; 95% CI 0.52, 0.86) in the univariable and (HR = 0.76; 95% CI 0.57, 1.01) in multivariable model, which also included age, (<40, ≥40 and <60, ≥60), sex, BMI group (<30, ≥30), BCG vaccination status (yes/no), Vitamin D deficiency, COVID-19 vaccination status (yes/no), and the presence of active TB. Also, in the multivariable model we observed a statistically significant effect of age at group (HR = 0.64; 95% CI 0.45, 0.91 and HR = 0.55; 95% CI 0.40, 0.77) in older age groups (<60 and ≥60), but the effect of Vitamin D deficiency was not significantly different (*p* = 0.10) between patients with high vs. low serum 25(OH)D levels (<10 ng/mL) (HR = 0.80; 95% CI 0.61, 1.05).

Of all patients who underwent QuantiFERON^®^-TB Gold (QFT-G), 59% tested negative, and 18% had tested positive signifying latent TB infection. Ten (91%) of the 11 patients who had active TB were hospitalized with severe or critical COVID-19, suggesting that they had a higher risk of falling into the two most severe categories (OR = 10.56; 95% CI 1.2, 93.0) (Table 2). The COPD increased the severity of the disease by 4.4 times (Table 2). The serum vitamin D levels did not differ between patients with active TB and COVID-19 inpatients vs. inpatients without active TB.

## 4. Discussion

Our study has shown that certain co-morbidities, such as having active TB and/or COPD, being male and of older age significantly affected the severity of COVID-19. Among 270 participants, 11 (4%) had active TB, 14 (5%) had a history of active TB, 39 (14%) had a household pulmonary TB contact, and 49 (18%) had tested QFT positive. Patients with active TB were more severely ill compared to those who did not have TB. On the other hand, patients with latent TB, tested by QFT, were less likely to develop severe COVID-19, but the difference was not statistically significant. Serum vitamin D levels did not have a significant effect on the association between active TB/LTBI and the severity of COVID-19. Active TB has been reported to influence the severity of COVID-19 in some studies [12] but the results have been mixed [13,14,15] with some studies reporting that latent TB infection was a protective factor against complications from COVID-19 [16]. Madan et al. conducted an observational study in 60 patients diagnosed with COVID-19. Among them, 15 tested positive for LTBI. The LTBI patients who had COVID-19 seemed to have milder symptoms [16]. Other researchers have also reported that LTBI induces lifelong innate immunity, leading to a better immune environment with a protective immunological response against COVID-19 with reduced severity and mortality [17,18,19,20]. On the other hand, patients with active TB disease who also contracted COVID-19 had a higher risk of severe disease and a poor outcome compared to those without TB [12,13,14,21,22].

In our study, a BCG vaccination scar was strongly protective against COVID-19 disease severity. Bacille Calmette-Guérin (BCG) is used to create a vaccine for tuberculosis. It is not widely used in the United States. BCG vaccination modulates both innate and acquired immunity and triggers a cascade of immunomodulation that is said to culminate in a heightened resilience against both viruses and bacteria [23] but there is ongoing debate concerning its actual effectiveness [24].

Serum 25(OH)D concentrations did not differ significantly between the patients whose illnesses were classified as mild/moderate compared to those whose illnesses were classified as severe/critical (*p* = 0.65). The current study was conducted during the coldest and darkest months in Mongolia when serum 25(OH)D levels have been shown be the lowest. Vitamin D deficiency (defined at <10 ng/mL) was found in 32% of our subjects, compared to 80% which we found in winter in a previous study [1]. Nevertheless, the mean levels of 25(OH)D in our current participants were only 17.9 ng/mL even though 69% (186 of the 270) were taking additional vitamin D supplementation. It is important to note, however, that acute systematic inflammatory illnesses like COVID-19 can lower serum 25(OH)D concentrations and reverse causality should be considered when evaluating the relationship between vitamin D and COVID-19 severity risk [25]. Cold temperature and dry conditions are strong predictors of COVID-19 severity due to effects on the diameter of exhaled aerosols containing SARS-CoV-2 and how long they stay airborne [26,27]. In addition, cold increases risk by cooling of the body surface and cold stress induced by lowering the core body temperature causes pathophysiological responses such as vasoconstriction in the respiratory tract mucosa and suppression of immune responses, which are responsible for increased susceptibility to infections [28,29]. In earlier studies, we reported that vitamin D may reduce the risk of both upper and lower respiratory tract infections including tuberculosis, and offers benefit particularly in people with vitamin D deficiency [30,31,32]. Some researchers have reported that COVID-19 patients supplemented with vitamin D were less likely to die, have an ICU admission, return to PCR positivity [33] and showed decreased pro-inflammatory cytokines and increased anti-inflammatory cytokines [34].

Our study has a number of strengths. We used the QuantiFERON test (as opposed to tuberculin skin test, TST) to detect *Mycobacterium tuberculosis* (MTB) infection which allowed MTB infection status to be evaluated without confounding by sensitization to BCG or environmental mycobacteria. We employed an objective assessment of Bacille Calmette–Guérin (BCG) status (presence vs. absence of BCG scar). The lab performing QuantiFERON tests participated in an External Quality Assurance scheme performed by an ISO 9001-accredited laboratory.

Our study also has some limitations. As with any observational study, associations observed may be due to residual and/or unmeasured confounding. However, the associations that we report are all biologically plausible and independent, withstanding adjustment for a wide range of potential confounders. We did not test for HIV infection; however, the prevalence of HIV infection in Mongolia is very low at 0.02% [35].

In conclusion, this analysis identifies active TB and COPD, as potentially modifiable risk factors for COVID-19 severity among Mongolian inpatients with vitamin D deficiency. The results of the current study call for an early evaluation of active TB in susceptible populations with COVID-19. This may mitigate against serious complications of COVID-19 infections or lessen the impact of acute respiratory distress syndrome (ARDS) in those who have been hospitalized.

## Figures and Tables

**Table 1 nutrients-15-03979-t001:** Characteristics of study participants (*n* = 270).

Characteristics		All Patients	Non-Severe	Severe
		N = 270	N = 125	N = 145
Sex	Female, *n* (%)	167 (62%)	84 (67%)	83 (57%)
	Male, *n* (%)	103 (38%)	41 (33%)	62 (43%)
Age, years, mean (sd)		56 (18)	52 (17)	59.5 (17)
Education ^1^	University/polytechnic, *n* (%)	121 (45%)	54 (43%)	67 (46%)
	Secondary school or lower, *n* (%)	128 (48%)	55 (44%)	73 (50%)
	NA	21 (8%)	16 (13%)	5 (3%)
Occupation	Unemployment	32 (12%)	16 (13%)	16 (11%)
	Salary employed	83 (31%)	46 (37%)	37 (26%)
	Self employed	40 (15%)	14 (11%)	26 (18%)
	Retired	115 (43%)	49 (39%)	66 (46%)
Type of residence	Centrally heated, *n* (%)	203 (75%)	93 (74%)	110 (76%)
	Not centrally heated, *n* (%)	38 (14%)	15 (12%)	23 (16%)
	Ger (Yurt), *n* (%)	29 (11%)	17 (14%)	12 (8%)
Patient actively smoking	Yes, *n* (%)	39 (14%)	19 (15%)	20 (14%)
Household PTB contact	Yes, *n* (%)	38 (14%)	15 (12%)	23 (16%)
BCG scar	Yes, *n* (%)	230 (85%)	123 (98%)	107 (74%)
Serum 25(OH)D, mean (sd)		18 (11)	17 (12)	18 (10)
Serum 25(OH)D^2^	<10 ng/mL, *n* (%)	86 (32%)	42 (34%)	44 (31%)
	≥10 ng/mL, *n* (%)	181 (68%)	83 (66%)	98 (69%)
Vitamin D supplementation use	Yes, *n* (%)	186 (69%)	86 (69%)	100 (69%)
Vitamin D supplementation dose	None	84 (31%)	39 (31%)	45 (31%)
	<1000	29 (101%)	9 (7%)	20 (14%)
	1000–2000	22 (8%)	8 (6%)	14 (10%)
	<2000	135 (50%)	69 (55%)	66 (46%)
BMI, kg/m^2^, mean (sd)		27 (5)	28 (6)	26 (5)
BMI, kg/m^2^	<30	207 (77%)	89 (71%)	118 (81%)
	≥30	63 (23%)	36 (29%)	27 (19%)
COVID-19 severity category (WHO)	Mild, *n* (%)	32 (12%)	32 (26%)	0 (0%)
	Moderate, *n* (%)	125 (46%)	93 (74%)	0 (0%)
	Severe, *n* (%)	86 (32%)	0 (0%)	86 (59%)
	Critical, *n* (%)	59 (22%)	0 (0%)	59 (41%)
Comorbidity	Any, *n* (%)	109 (40%)	54 (43%)	55 (38%)
	Diabetes, *n* (%)	38 (14%)	16 (13%)	22 (15%)
	Hypertension, *n* (%)	128 (47%)	63 (50%)	65 (45%)
	COPD, *n* (%)	36 (13%)	8 (6%)	28 (19%)
	Heart Attack/Stroke, *n* (%)	13 (5%)	5 (4%)	8 (6%)
	Heart Failure, *n* (%)	3 (1%)	0 (0%)	3 (2%)
	Bypass, *n* (%)	12 (4%)	4 (3%)	8 (6%)
	Sleep Apnea, *n* (%)	54 (20%)	16 (13%)	38 (26%)
	Kidney Stone, *n* (%)	5 (2%)	3 (2%)	2 (1%)
	Kidney Failure, *n* (%)	11 (4%)	4 (3%)	7 (5%)
	Chronic liver disease, *n* (%)	16 (6%)	9 (7%)	7 (5%)
	Hypercalcemia, *n* (%)	1 (0.4%)	0 (0%)	1 (1%)
	Parathyroid disease, *n* (%)	38 (14%)	3 (2%)	12 (8%)
	Sarcoidosis, *n* (%)	9 (3%)	1 (1%)	8 (6%)
	TB, *n* (%)	11 (4%)	1 (1%)	10 (7%)
	Advanced cancer, *n* (%)	13 (5%)	7 (6%)	6 (4%)
Vaccination doses ^2^	No	24 (9%)	5 (4%)	19 (13%)
	I dose	20 (7%)	3 (2%)	17 (12%)
	Full dose	101 (37%)	63 (50%)	38 (26%)
	Boosting (III) doses	109 (40%)	46 (37%)	63 (43%)
	Boosting (IV) doses	16 (6%)	8 (6%)	8 (6%)
QFT-G	Negative	159 (59%)	72 (58%)	87 (60%)
	Positive	49 (18%)	26 (21%)	23 (16%)
	Indetermined	5 (2%)	1 (1%)	4 (3%)
	Other	57 (21%)	26 (21%)	31 (21%)
Fever	None	177 (66%)	83 (66%)	94 (65%)
	Mild, *n* (%)	55 (20%)	13 (10%)	42 (29%)
	Moderate, *n* (%)	35 (13%)	27 (22%)	8 (6%)
	Severe, *n* (%)	3 (1%)	2 (2%)	1 (1%)
Cough	None	14 (5%)	7 (6%)	7 (5%)
	Mild, *n* (%)	141 (52%)	28 (22%)	113 (78%)
	Moderate, *n* (%)	85 (32%)	69 (55%)	16 (11%)
	Severe, *n* (%)	30 (11%)	21 (17%)	9 (6%)
Sore throat	None	104 (39%)	46 (37%)	58 (40%)
	Mild, *n* (%)	105 (39%)	32 (26%)	73 (50%)
	Moderate, *n* (%)	50 (19%)	39 (31%)	11 (8%)
	Severe, *n* (%)	11 (4%)	8 (6%)	3 (2%)
Stuffy or runny nose	None	180 (67%)	87 (70%)	93 (64%)
	Mild, *n* (%)	70 (26%)	21 (17%)	49 (34%)
	Moderate, *n* (%)	15 (6%)	13 (10%)	2 (1%)
	Severe, *n* (%)	5 (2%)	4 (3%)	1 (1%)
Chest pain	None	87 (32%)	43 (34%)	44 (30%)
	Mild, *n* (%)	102 (38%)	23 (18%)	79 (55%)
	Moderate, *n* (%)	59 (22%)	44 (35%)	15 (10%)
	Severe, *n* (%)	22 (8%)	15 (12%)	7 (5%)
Headache	None	70 (26%)	36 (29%)	34 (23%)
	Mild, *n* (%)	118 (48%)	33 (26%)	85 (59%)
	Moderate, *n* (%)	63 (23%)	43 (34%)	20 (14%)
	Severe, *n* (%)	19 (7%)	13 (10%)	6 (4%)
Fatigue	None	32 (12%)	12 (10%)	20 (14%)
	Mild, *n* (%)	99 (37%)	31 (25%)	68 (47%)
	Moderate, *n* (%)	90 (33%)	52 (42%)	38 (26%)
	Severe, *n* (%)	49 (18%)	30 (24%)	19 (13%)
Nausea	None	189 (70%)	91 (73%)	98 (68%)
	Mild, *n* (%)	55 (20%)	22 (18%)	33 (23%)
	Moderate, *n* (%)	23 (9%)	10 (8%)	13 (9%)
	Severe, *n* (%)	3 (1%)	2 (2%)	1 (1%)
Diarrhea	None	250 (93%)	119 (95%)	131 (90%)
	Mild, *n* (%)	17 (6%)	4 (3%)	13 (9%)
	Moderate, *n* (%)	2 (0.7%)	1 (1%)	1 (1%)
	Severe, *n* (%)	1 (0.4%)	1 (1%)	0 (0%)
Sense of taste	Usual, *n* (%)	41 (15%)	28 (22%)	13 (9%)
	Less than usual, *n* (%)	33 (12%)	16 (13%)	17 (12%)
	No sense of taste, *n* (%)	196 (73%)	81 (65%)	115 (79%)
Sense of smell	Usual, *n* (%)	46 (17%)	31 (25%)	15 (10%)
	Less than usual, *n* (%)	26 (10%)	12 (10%)	14 (10%)
	No sense of taste, *n* (%)	198 (74%)	82 (66%)	116 (80%)

Abbreviations: BCG, Bacille Calmette–Guérin vaccine; PTB, pulmonary TB; BMI, body mass index; QFT-G, QuantiFERON^®^-TB Gold; sd, standard deviation; ^1^ Highest educational level attained by either parent. ^2^ Data missing for *n* = 3.

**Table 2 nutrients-15-03979-t002:** Risk factors for severe or critical COVID-19 disease among hospitalized patients with COVID-19.

		Proportion of Severe (%)	Univariable Model		Multivariable Model	
Variables			OR (95% CI)	*p*-Value	OR (95% CI)	*p*-Value
Gender	Female	83 (50)	Ref			
	Male	62 (60)	1.53 (0.93, 2.52)	0.09	2.15 (1.17, 3.94)	0.01
Age	<40	22 (39)	Ref		Ref	
	40–60	47 (52)	1.69 (0.86, 3.33)	0.13	1.45 (0.63, 3.34)	0.38
	>60	76 (61)	2.45 (1.28, 4.67)	0.007	2.56 (1.17, 5.63)	0.02
Occupation	Salary employed	16 (50)	Ref			
	Self employed	37 (45)	0.8 (0.36, 1.82)	0.6		
	Retired	26 (65)	1.86 (0.72, 4.8)	0.2		
	Unemployed	66 (57)	1.35 (0.61, 2.95)	0.46		
Education	University/polytechnic	61 (60)	Ref			
	Secondary/lower	79 (53)	0.75 (0.45, 1.25)	0.27		
Type of residence	Centrally heated	110 (54)	Ref			
	Not centrally heated	23 (61)	1.3 (0.64, 2.63)	0.47		
	Ger (Yurt)	12 (41)	0.6 (0.27, 1.31)	0.2		
Smoking	No	125 (54)	Ref			
	Yes	20 (51)	0.89 (0.45, 1.76)	0.74		
Alcohol consumption	No	127 (52)	Ref			
	Yes	18 (69)	2.07 (0.87, 4.95)	0.1		
QFT-G	Negative	87 (55)	Ref			
	Positive	23 (47)	0.73 (0.39, 1.39)	0.34		
	Indetermined	4 (80)	3.31 (0.36, 30.25)	0.29		
BCG	No	38 (95)	Ref			
	Yes	107 (47)	0.05 (0.01, 0.19)	<0.001	0.04 (0.01, 0.16)	<0.001
Serum 25(OH)D	≥10 ng/mL, n (%)	98 (54)	Ref			
	<10 ng/mL, n (%)	44 (51)	0.89 (0.53, 1.48)	0.65		
Vitamin D Supplementation	Yes	45 (54)	Ref			
	No	100 (54)	1.01 (0.6, 1.69)	1		
BMI, kg/m^2^	<30	118 (57)	Ref			
	≥30	27 (43)	0.57 (0.32, 1)	0.05	0.57 (0.27, 1.2)	0.14
COVID-19 vaccination	Yes	19 (79)	Ref		Ref	
	No	126 (51)	0.28 (0.1, 0.76)	0.01	0.26 (0.08, 0.85)	0.03
Comorbidity	Yes	90 (56)	Ref			
	No	55 (51)	0.8 (0.49, 1.31)	0.38		
Active tuberculosis	No	135 (52)	Ref			
	Yes	10 (91)	9.19 (1.16, 72.77)	0.04	10.56 (1.2, 93)	0.03
Diabetes	No	123 (53)	Ref			
	Yes	22 (58)	1.22 (0.61, 2.44)	0.58		
Hypertension history	No	80 (56)	Ref			
	Yes	65 (51)	0.8 (0.49, 1.29)	0.36		
COPD	No	117 (50)	Ref		Ref	
	Yes	28 (78)	3.5 (1.53, 8)	0.003	4.41 (1.76, 11.09)	0.002
History of heart attack	No	137 (53)	Ref			
	Yes	8 (62)	1.4 (0.45, 4.4)	0.56		
History of coronary bypass	No	137 (53)	Ref			
	Yes	8 (67)	1.77 (0.52, 6.01)	0.36		
Diagnosed as sleep apnea	No	107 (50)	Ref			
	Yes	38 (70)	1.22 (0.61, 2.44)	0.58		
Kidney failure or dialysis	No	138 (53)	Ref			
	Yes	7 (64)	1.53 (0.44, 5.37)	0.5		
Severe liver disease or cirrhosis	No	138 (54)	Ref			
	Yes	7 (44)	0.65 (0.24, 1.81)	0.41		
Parathyroid	No	133 (52)	Ref			
	Yes	12 (80)	3.67 (1.01, 13.31)	0.05	4.35 (0.96, 19.8)	0.06
Sarcoid	No	137 (53)	Ref			
	Yes	8 (89)	7.24 (0.89, 58.72)	0.06	4.56 (0.46, 45.45)	0.2
Advanced cancer	No	139 (54)	Ref			
	Yes	6 (46)	1.22 (0.61, 2.44)	0.58		
Antiviral drug use	Yes	135 (57)	Ref			
	No	10 (31)	0.35 (0.16, 0.76)	0.009	0.55 (0.22, 1.36)	0.19
Corticosteroid use	Yes	64 (48)	Ref		Ref	
	No	81 (59)	1.56 (0.96, 2.52)	0.07	1.55 (0.85, 2.81)	0.15
NSAID	Yes	86 (53)	Ref			
	No	59 (55)	1.1 (0.67, 1.8)	0.7		
ACE inhibitors	Yes	95 (55)	Ref			
	No	50 (51)	0.84 (0.51, 1.39)	0.51		
ARBs	Yes	118 (57)	Ref			
	No	27 (42)	0.54 (0.31, 0.96)	0.04	0.57 (0.28, 1.18)	0.13

Abbreviations: QFT-G, QuantiFERON^®^-TB Gold; COPD, chronic obstructive pulmonary disease; NSAID, Non-steroidal anti-inflammatory drugs; ACE, angiotensin-converting enzyme; ARB, Angiotensin receptor blockers. Odds ratios for factors associated with COVID-19 severity and their 95% confidence intervals were estimated using logistics regression.

**Table 3 nutrients-15-03979-t003:** Laboratory parameters for severe or critical COVID-19 disease among hospitalized patients with COVID-19.

Variables	Mean (sd) in Severe Group	Univariable Model		Multivariable Model	
		OR (95% CI)	*p*-Value	OR (95% CI)	*p*-Value
LMR	8 (37)	0.99 (0.99, 1.01)	0.72		
Fibrinogen (g/L)	3.9 (1.2)	0.96 (1.47, 1.47)	0.11		
aPTT (s)	45 (28)	1 (1.06, 1.06)	0.09	1.02 (0.98, 1.06)	0.31
TT (s)	20.5 (7.4)	1.06 (1.24, 1.24)	0.001	1.17 (1.06, 1.29)	0.001
INR	1.16 (1)	0.55 (4.09, 4.09)	0.43		
PT (s)	13.9 (3.7)	0.9 (1, 1)	0.06	0.94 (0.88, 1.01)	0.1
WBC (10^3^/µL)	6.9 (4.8)	0.97 (1.08, 1.08)	0.31		
HCT (%)	36.2 (9.2)	0.91 (0.98, 0.98)	0.002	0.97 (0.93, 1.01)	0.10
NEUT (10^3^/µL)	5.7 (10.4)	0.99 (1.08, 1.08)	0.11		
LYMPH (10^3^/µL)	2.3 (3)	0.95 (1.11, 1.11)	0.56		
MON (10^3^/µL)	0.7 (1.3)	0.91 (2.02, 2.02)	0.13		
Ferritin (ng/mL)	392.8 (420.9)	1 (1, 1)	0.47		
CRP (ng/mL)	27,291.2 (28,792.7)	1 (1, 1)	<0.001	1 (1, 1)	<0.001
IL-6 (pg/mL)	30 (260)	1 (1.03, 1.03)	0.07	1.01 (0.99, 1.02)	0.42
D-Dimer (mu g/mL)	0.5 (0.2)	6.8 (175.1, 175.1)	<0.001	1.07 (0.82, 1.38)	0.63
PCT (ng/mL)	0.3 (0.2)	0.34 (2.57, 2.57)	0.89		

Abbreviations: LMR, lymphocyte and monocyte ratio; aPTT, activated partial thromboplastin clotting time; TT, thrombin time (TT); INR, international normalized ratio; PT, prothrombin time; WBC, white blood cells; HCT, hematocrit, NEUT, neutrophils; LYMPH, lymphocytes; MON, monocytes; CRP, C-reactive protein; IL-6, interleikin-6; PCT, procalcitonin.

## Data Availability

Study data set is available from the corresponding author upon reasonable request.

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
