# Peer review of "Latent TB Infection, Vitamin D Status and COVID-19 Severity in Mongolian Patients"

_nutrients, 2023, doi:10.3390/nu15183979_

Round 1
Reviewer 1 Report
This manuscript has interesting results which should have clinical relevance in countries with high TB rates.
It should be discussed that having an acute inflammatory illness such as COVID-19 lowers serum 25(OH)D concentrations. Therefore, the values measured at time of hospitalization can not be used to evaluate risk of developing COVID-19 without considering how much they were lowered due to having COVID-19, or at least noting the problem.
Letter to the Editor: Vitamin D deficiency in COVID-19: Mixing up cause and consequence.
Smolders J, van den Ouweland J, Geven C, Pickkers P, Kox M.Metabolism. 2021 Feb;115:154434. doi: 10.1016/j.metabol.2020.154434.
This letter has 68 citations listed at Google Scholar. Some of them might be cited as well.
It could be noted that temperature and humidity are strong predictors of COVID-19 due to effects on the diameter of exhaled aerosols containing SARS-CoV-2 and how long they stay airborne.
Temperature, humidity, and latitude analysis to predict potential spread and seasonality for COVID-19
MM Sajadi, P Habibzadeh, A Vintzileos… - Social Science …, 2020 - ncbi.nlm.nih.gov
Effects of temperature and humidity on the spread of COVID-19: A systematic review
P Mecenas, RTRM Bastos, ACR Vallinoto… - PLoS one, 2020 - journals.plos.org
In addition, cold increases risk this way
cooling of the body surface and cold stress induced by lowering the core body temperature cause pathophysiological responses such as vasoconstriction in the respiratory tract mucosa and suppression of immune responses, which are responsible for increased susceptibility to infections.
Exposure to cold and respiratory tract infections.
Int J Tuberc Lung Dis. 2007 Sep;11(9):938-43.
Exposure to cold and acute upper respiratory tract infection.
Rhinology. 2015 Jun;53(2):99-106. doi: 10.4193/Rhino14.239.
In our study BCG vaccination scar was strongly protective against COVID-19 disease severity. BCG vaccination
Comment: Suggest defining BCG if some of the readers may not know what it stands for.
Bacille Calmette-Guérin (BCG) is a vaccine for tuberculosis (TB) disease. This vaccine is not widely used in the United States
|
LMR |
8.23 (36.5) |
0.99 (1.01,1.01) |
The 95% CI values in Table 3 for the univariate analysis are incorrect as the low value is the high value repeated.
References: The references are not in the preferred Nutrients format and page numbers are missing on several. The initials precede the last name.
[20] N. Gupta, P. Ish, A. Gupta, N. Malhotra, J.A. Caminero, R. Singla, R. Kumar, S.R. Yadav, N. Dev, S. Agrawal, S. Kohli, M.K. Sen, S. Chakrabarti, N.K. Gupta, A profile of a retrospective cohort of 22 patients with COVID-19 and active/treated tuberculosis, The European respiratory journal, 56 (2020).
From a previous Nutrients article by the same authors – initials follow the last name.
1. Ganmaa, D.; Holick, M.F.; Rich-Edwards, J.W.; Frazier, L.A.; Davaalkham, D.; Ninjin, B.; Janes, C.; Hoover, R.N.; Troisi, R. Vitamin D deficiency in reproductive age Mongolian women: A cross sectional study. J. Steroid Biochem. Mol. Biol. 2014, 139, 1–6. [Google Scholar] [CrossRef][Green Version]
Please be sure to use the MDPI Endnote template.
Significant digits. The general rule is that no more non-zero digits should be given than are justified by the uncertainty of the value.
See "Too many digits: the presentation of numerical data"
https://www.ncbi.nlm.nih.gov/pmc/articles/PMC4483789/
If the uncertainty (or difference when comparing numbers) is greater than about 7%, only two non-zero digits are justified.
P values should be given to two decimal places unless the first two are 00 or the number lies between 0.045 and 0.054. If the first two are 00, then only one non-zero digit can be given.
Thus, p values should be adjusted.
|
LMR |
8.23 (36.5) |
|
Should be |
8 (37) |
|
Fibrinogen (g/l) |
3.87 (1.2) |
Should be
|
Fibrinogen (g/l) |
3.9 (1.2) |
|
APTT (sec) |
44.89 (28.1) |
Should be
|
APTT (sec) |
45 (28) |
Percentages would be better reported in whole numbers. There is no need to know the decimal value, and whole numbers are easier to grasp quickly.
Please review all numbers in abstract, text, tables, and figures and adjust accordingly.
Author Response
|
Thank you very much for taking the time to review this manuscript. I am glad that you found that our manuscript has interesting results and have clinical relevance in countries with high TB rates. Please see the detailed responses below and the corresponding revisions/corrections highlighted in the re-submitted files. |
|
Comments 1: It should be discussed that having an acute inflammatory illness such as COVID-19 lowers serum 25(OH)D concentrations. Therefore, the values measured at time of hospitalization can not be used to evaluate risk of developing COVID-19 without considering how much they were lowered due to having COVID-19, or at least noting the problem. Smolders J, van den Ouweland J, Geven C, Pickkers P, Kox M.Metabolism. 2021 Feb;115:154434. doi: 10.1016/j.metabol.2020.154434. This letter has 68 citations listed at Google Scholar. Some of them might be cited as well. It could be noted that temperature and humidity are strong predictors of COVID-19 due to effects on the diameter of exhaled aerosols containing SARS-CoV-2 and how long they stay airborne. Temperature, humidity, and latitude analysis to predict potential spread and seasonality for COVID-19 MM Sajadi, P Habibzadeh, A Vintzileos… - Social Science …, 2020 - ncbi.nlm.nih.gov Effects of temperature and humidity on the spread of COVID-19: A systematic review P Mecenas, RTRM Bastos, ACR Vallinoto… - PLoS one, 2020 - journals.plos.org |
|
Response 1: Thank you very much for your helpful comments and providing these references. We agree with your comments. We have incorporated your comments into the Discussion, page 9, and have added the relevant references accordingly |
|
Comments 2: Suggest defining BCG if some of the readers may not know what it stands for. Bacille Calmette-Guérin (BCG) is a vaccine for tuberculosis (TB) disease. This vaccine is not widely used in the United States |
|
Response 2: Agree. We have, accordingly, added the definition and your comment into the Discussion, page 9 and highlighted the text. Comments 3: The 95% CI values in Table 3 for the univariate analysis are incorrect as the low value is the high value repeated. Response 3: Thank you! We have corrected this accordingly in Table 3. Comments 4, References: The references are not in the preferred Nutrients format and page numbers are missing on several. The initials precede the last name. Response 4: Thank you. We have revised the references accordingly. We have used the MDPI Endnote template in the revised manuscript.
|
Reviewer 2 Report
This is a very interesting noble study with appropriate experiments. The interpretation is good with proper language and claims. The overall rigor and relevance to Nutrients is very strong. I am supportive of this paper however a few minor aspects below will need to be clarified before it is considered for publication.
The methods can be written in detail.
The results look promising, and the biological interpretation of the results seems okay.
Please include a list of abbreviations and their full meanings for ease of reference.
The discussion can be made both more streamlined and effective. Some of it reads as a restatement of results, which is not really necessary. It would, however, benefit from a more mechanistic explanation or even.
Minor typos/English check.
Author Response
|
Thank you very much for taking the time to review this manuscript. |
I am glad that you found our study to be “very interesting noble study with appropriate experiments” Please see below for our responses.
Comment 1: Please include a list of abbreviations and their full meanings for ease of reference.
Response 1. I have included a list of abbreviations and their full meanings in the revised manuscript, page 1, in highlighted text.
Comment 2: The discussion can be made both more streamlined and effective. Some of it reads as a restatement of results, which is not really necessary. It would, however, benefit from a more mechanistic explanation or even.
Response 2. Thank you. We have revised the Discussion section accordingly and have added more explanations.
Comments on the Quality of English Language
Comment 3: Minor typos/English check.
Response 2. Thank you. We have edited the language accordingly.